 eLIFE

# A sleep state in *Drosophila* larvae required for neural stem cell proliferation

**Milan Szuperak[1], Matthew A Churgin[2], Austin J Borja[1], David M Raizen[3,4,5], Christopher Fang-Yen[2,6], Matthew S Kayser[1,4,5,6]***

[1]Department of Psychiatry, Perelman School of Medicine at the University of Pennsylvania, Philadelphia, United States; [2]Department of Bioengineering, School of Engineering and Applied Science, University of Pennsylvania, Philadelphia, United States; [3]Department of Neurology, Perelman School of Medicine at the University of Pennsylvania, Philadelphia, United States; [4]Chronobiology Program, Perelman School of Medicine at the University of Pennsylvania, Philadelphia, United States; [5]Center for Sleep and Circadian Neurobiology, Perelman School of Medicine at the University of Pennsylvania, Philadelphia, United States; [6]Department of Neuroscience, Perelman School of Medicine at the University of Pennsylvania, Philadelphia, United States

**Abstract** Sleep during development is involved in refining brain circuitry, but a role for sleep in the earliest periods of nervous system elaboration, when neurons are first being born, has not been explored. Here we identify a sleep state in *Drosophila* larvae that coincides with a major wave of neurogenesis. Mechanisms controlling larval sleep are partially distinct from adult sleep: octopamine, the *Drosophila* analog of mammalian norepinephrine, is the major arousal neuromodulator in larvae, but dopamine is not required. Using real-time behavioral monitoring in a closed-loop sleep deprivation system, we find that sleep loss in larvae impairs cell division of neural progenitors. This work establishes a system uniquely suited for studying sleep during nascent periods, and demonstrates that sleep in early life regulates neural stem cell proliferation.
DOI: https://doi.org/10.7554/eLife.33220.001

*For correspondence:
kayser@pennmedicine.upenn.edu

**Competing interests:** The authors declare that no competing interests exist.

## Introduction

Nearly all animals exhibit increased sleep throughout development, suggesting a crucial role for sleep in early life (*Kayser and Biron, 2016*; *Roffwarg et al., 1966*). Many studies have focused on the role of sleep in synaptic formation, homeostasis, and plasticity (*Abel et al., 2013*; *Frank, 2011*; *Kayser et al., 2014*; *Li et al., 2017*; *Tononi and Cirelli, 2014*), underscoring that a core function of sleep during development is likely to involve modification of neural connectivity. However, examination of a function for sleep in earlier phases of brain development has been limited by the lack of a tractable experimental system. Indeed, there is no established platform to investigate the role of sleep in developmental neurogenesis, which in mammals predominately occurs *in utero*. The adult fruit fly is a widely studied model organism for sleep (*Hendricks et al., 2000*; *Shaw et al., 2000*), and while significant brain plasticity is ongoing in young adult flies (*Barth et al., 1997*), neurogenesis is largely complete; adult flies are therefore not suitable for investigating the function of sleep during initial periods of neural development. In contrast, the timing of these events in *Drosophila* larvae is ideal. The first wave of neurogenesis occurs during embryonic stages but contributes only 10% of neurons in the adult brain; 90% of neurons in the adult brain are born during the second wave of neurogenesis that is timed to larval stages (*Homem and Knoblich, 2012*; *Kohwi and Doe, 2013*). Fly larvae are widely used to study the neural basis of complex behaviors, and are readily accessible

**eLife digest** Nearly all animals sleep more while they are still developing, suggesting that sleep is important in early life. Previous studies have shown that sleep may be required for building connections in the brain. However, it has been difficult to study the effects of sleep in earlier stages of brain development, when stem cells divide to create brain cells in a process known as "neurogenesis". This is partly because, in mammals, most neurogenesis occurs in the womb.

Scientists have successfully studied sleep using the common fruit fly. But these studies have so far focused on adult flies, in which neurogenesis is mostly complete. Fly larvae, on the other hand, are widely used to study brain development and neurogenesis. Compared to mammals in the womb, fruit fly larvae are very easy to access and manipulate. However, unlike adult flies, no one had previously looked to see if larvae even display a behaviour that would fit the definition of sleep.

To see if fruit fly larvae do sleep, Szuperak et al. created the "LarvaLodge", an apparatus in which individual larvae can be housed while having their activity monitored over time. In these lodges, a bright light was used to test how hard it is to arouse inactive fruit fly larvae, and further experiments asked what happens when larvae are prevented from resting. Then, to look at neurogenesis in the larvae, Szuperak et al. used a stain that labels dividing stem cells within the nervous system. Those cells could then be seen and counted when a larva was dissected and examined under a microscope.

The results from the LarvaLodge showed that fruit fly larvae do indeed sleep: they have extended periods of rest during which they react less to outside disturbances and adopt a particular posture (they retract their heads towards their bodies). Also when larvae were deprived of sleep, by shining a light or shaking, they compensated by sleeping more afterwards. Importantly, depriving the larvae of sleep also led to lower levels of neurogenesis. These findings establish the fruit fly larva as a new and useful system for studying the role of sleep in early development, and may help shed light on the role sleep plays in disorders affecting brain development.

DOI: https://doi.org/10.7554/eLife.33220.002

for experimental manipulation (*Heckscher et al., 2012*; *Honjo et al., 2012*; *Luo et al., 2010*; *Vogelstein et al., 2014*). Yet, it has remained unknown whether *Drosophila* larvae sleep. Here, we identify a sleep state in *Drosophila* larvae, define the principal cellular controls of larval sleep, and provide evidence that sleep during early development is required for normal proliferation of neural stem cells. Our results suggest that sleep is a critical factor in the initial steps of nervous system elaboration.

## Results

### Prolonged monitoring of larval rest-activity behaviors

Most prior research on *Drosophila* larval behavior (*Heckscher et al., 2012*; *Vogelstein et al., 2014*; *Widmann et al., 2016*) employed short-term assays not sufficient for analysis of rest/activity. Building on work in *Caenorhabditis elegans* (*Churgin et al., 2017*; *Raizen et al., 2008*), we constructed a multi-well imaging device, the LarvaLodge, for automated, prolonged monitoring of larval activity and quiescence (*Figure 1A–C*). We simultaneously imaged up to 20 larvae at 35 μm resolution (a second instar larva is ~1500 μm in length and ~350 μm in width), acquiring images every 6 s under dark field infrared illumination (*Video 1*). We focused on second instar larvae because this period coincides with the second wave of neurogenesis (*Homem and Knoblich, 2012*). With this system, rest/activity during the entire second instar phase (~22 hr) was recorded, with tightly aligned developmental staging across larvae as indicated by synchronized onset of the second molt (*Figure 1C*). We observed that larvae exhibit dynamic transitions between quiescence and activity throughout the imaging period (*Figure 1C*; *Video 2*). Our recordings of activity were sensitive to both locomotor and feeding behaviors (*Figure 1—figure supplement 1A*; *Video 3*), demonstrating our ability to distinguish behavioral quiescence from eating. Total quiescence was higher early in the second instar period compared to later, resulting from more frequent and longer sleep bouts in immature larvae (*Figure 1D–F*; *Figure 1—figure supplement 1B,C*). These results demonstrate our ability to monitor

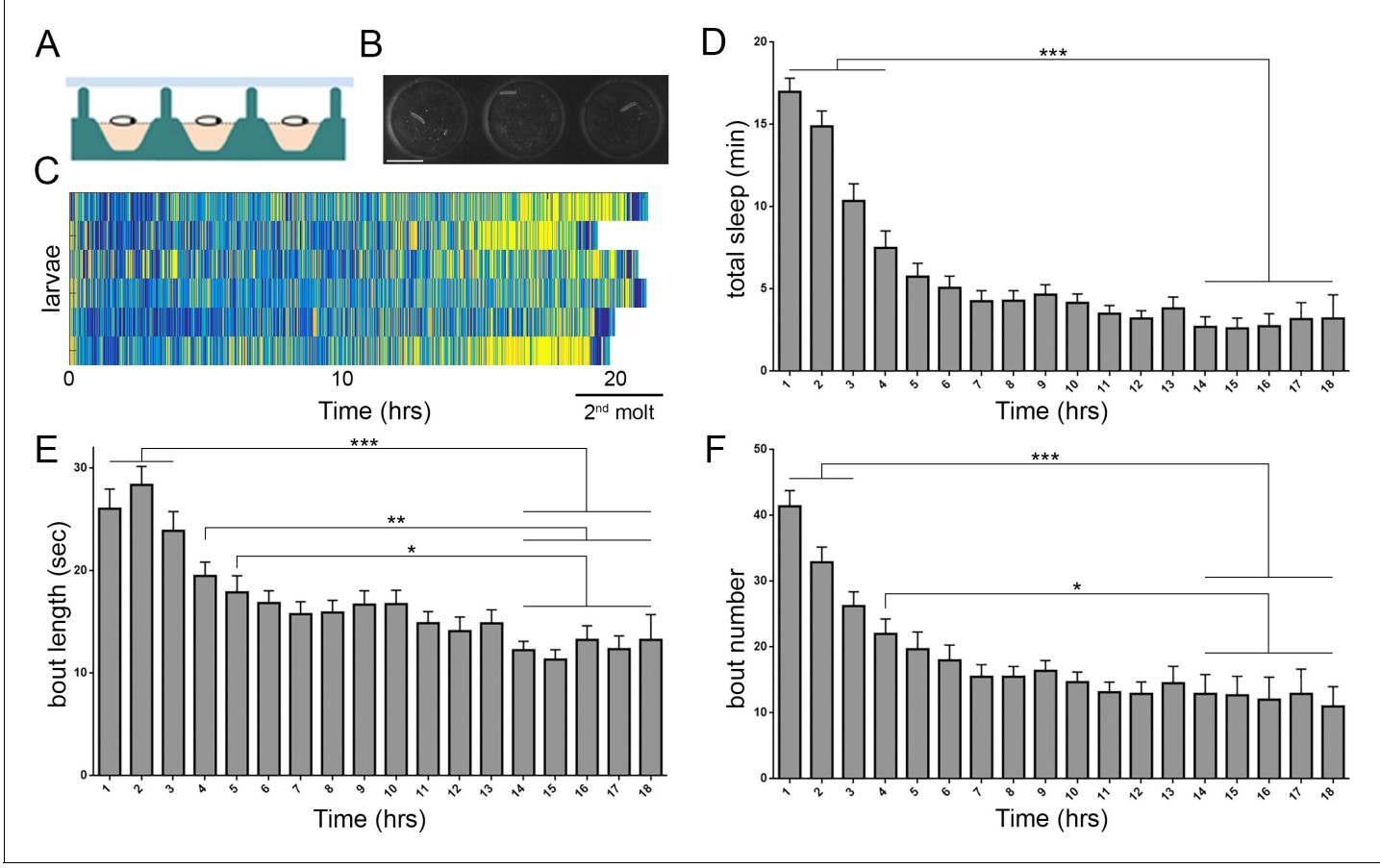

**Figure 1.** Prolonged monitoring of larval behavior in LarvaLodge devices. (**A**) Schematic of a cross section through the device. PDMS (silicone) is in green, agar in pink, and ceiling above in light blue. (**B**) Image of 3 LarvaLodge wells containing individual second instar larvae. (**C**) Heat map of activity of 6 larvae monitored from early second instar through second molt (blue = inactive, yellow = high activity). Quantification of total quiescence (**D**), quiescence bout length (**E**), and quiescence bout number (**F**) in hourly bins throughout the second instar stage (n = 20 larvae). In this and other figures, *p≤0.05; **p<0.01; ***p<0.001; error bars = SEM. Scale bar = 5 μm.

DOI: https://doi.org/10.7554/eLife.33220.003

The following figure supplement is available for figure 1:

**Figure supplement 1.** Detection of feeding and temporal distribution of quiescence in LarvaLodges.

DOI: https://doi.org/10.7554/eLife.33220.004

larval activity over extended periods, and reveal behavioral quiescence that is developmentally timed.

## Larval behavioral quiescence is a sleep state

All animals appear to sleep, defined behaviorally as a state of quiescence that is rapidly reversible, associated with a postural change, and during which arousal threshold is increased; sleep loss is followed by homeostatic sleep rebound (**Allada and Siegel, 2008**; **Campbell and Tobler, 1984**). We tested whether behavioral quiescence in larvae represents a sleep state by examining these behavioral criteria. First, we asked if larval quiescence is associated with a postural change. Using frame-by-frame postural analysis of individual sleep bouts, we observed that during periods of rest, larvae frequently retract their head towards the body, resulting in a quantitative change in body length and width (**Figure 2A–E**; **Video 4**). Head retraction occurred in 76.6% of prolonged quiescence bouts (>36 s; **Figure 2D**) and less commonly (47.1%) in shorter bouts (18 s; **Figure 2C**), suggesting that postural change is related to duration of the sleep bout. The postural change most often occurred as part of a stereotyped sequence of behaviors: we found in 72.0% of sleep-associated postural changes that the animal transitioned from activity to behavioral quiescence prior to postural change;

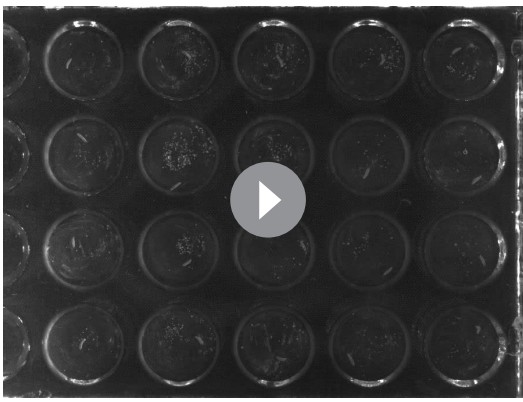

**Video 1.** Time-lapse video of twenty second instar larvae in the LarvaLodge. For this and all videos, frames are acquired every 6 s and videos are shown at six frames per second (36x faster than real time).
DOI: https://doi.org/10.7554/eLife.33220.005

quiescence then persisted following postural change in 64.6% of episodes, prior to resumption of activity (*Video 4*). Less often (28.0%) a larva transitioned directly from activity to coincident postural change with quiescence, or directly from head retraction to an active phase (35.4%), consistent with the observation that posture is more likely to change in association with prolonged rest. Our data thus show that *Drosophila* larval quiescence is a patterned behavior with a previously unrecognized temporal sequence, and that like animals ranging from nematodes (*Iwanir et al., 2013*) to humans, larvae demonstrate postural changes during sleep.

To determine whether larval rest is a rapidly reversible behavioral state, we randomly presented a high intensity blue light stimulus (irradiance 39.8 µW/mm$^2$) for 4 s to second instar larvae and retrospectively analyzed behavioral data for larvae that were inactive at the time of stimulus presentation. We found that nearly 100% of inactive larvae were aroused (as determined by altered position in next captured frame) by this intense stimulus (*Figure 2F*; *Video 5*), indicating that behavioral quiescence is rapidly reversible. Using a low intensity blue light stimulus (irradiance 3.98 µW/mm$^2$) to perturb larvae, we found that only ~60% of inactive larvae became active (*Figure 2F*), demonstrating an altered responsiveness to sensory stimuli during larval rest. Importantly, only ~20% of quiescent larvae spontaneously became active without a stimulus (*Figure 2F*), confirming that the low intensity stimulus arouses resting larvae more than chance. We next increased the temporal resolution of our system to monitor behavior every 2 s, with the goal of examining whether the behavioral quiescence described above is

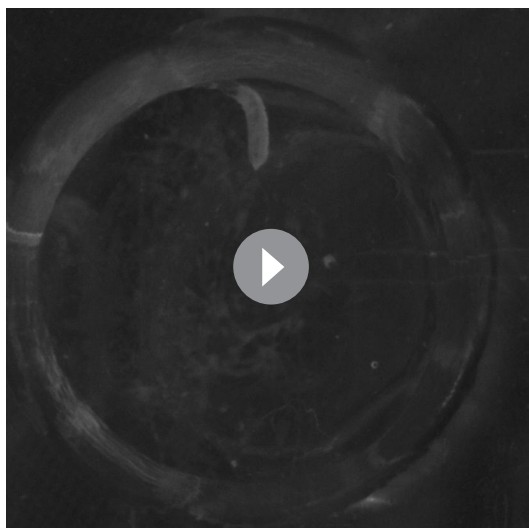

**Video 2.** Time-lapse (36x) of a second instar larva exhibiting transitions between activity and quiescence. Quiescence bouts occur at 00:07 and 00:15. The quiescent bout occurring from 00:07 to ~ 00:12 lasts ~ 180 s.
DOI: https://doi.org/10.7554/eLife.33220.006

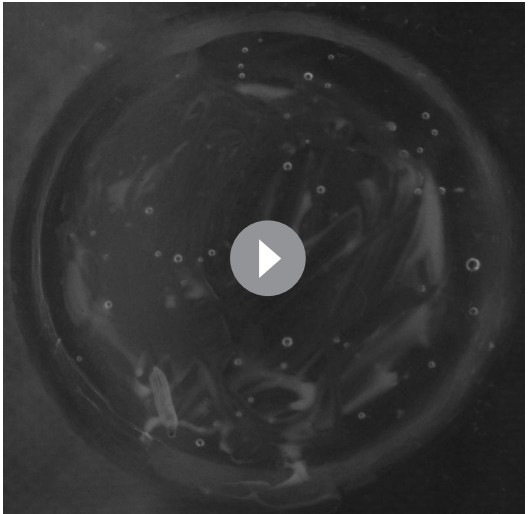

**Video 3.** Time-lapse (36x) of a second instar larva demonstrating feeding behaviors, which are detected as activity in our system. Feeding begins at time = 00:12 and is characterized by pumping and sweeping head motions, during which the underlying food begins to disappear.
DOI: https://doi.org/10.7554/eLife.33220.007

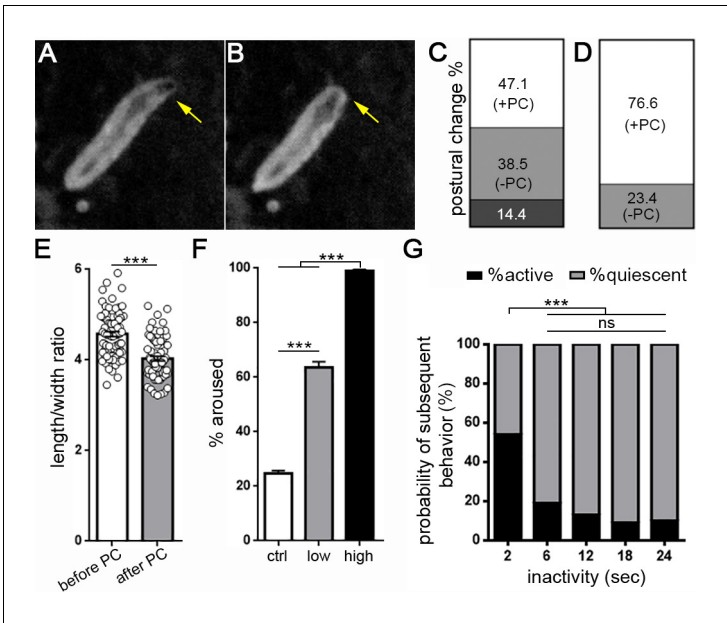

**Figure 2.** *Drosophila* larval quiescence meets behavioral criteria for sleep. Image of a larva before (**A**) and after (**B**) postural change associated with sleep. Yellow arrow indicates head retraction. Quantification of postural change frequency associated with 18 s sleep bout length (**C**) (n = 104 sleep episodes) or ≥36 s sleep bout length (**D**) (n = 107 sleep episodes) (white=% sleep episodes with postural change (+PC); light gray = no postural change (-PC); dark gray = not determined). (**E**) Quantification of larval length/width ratio before and after postural change. (**F**) Percentage of larvae aroused from quiescence using a high intensity (black bar, n = 187 sleep episodes) or low intensity stimulus (gray, n = 119 sleep episodes).~20% of larvae wake spontaneously in absence of a stimulus (white, n = 312). (**G**) Probability of spontaneous activity (black) or continued quiescence (gray) following a defined period of inactivity (n = 274, 88, 53, 59, 52 quiescent episodes from left to right).
DOI: https://doi.org/10.7554/eLife.33220.008

qualitatively distinct from brief (2 s) periods of inactivity. Focusing first on probability of spontaneous arousal, we found that following 2 s of inactivity, 54% of larvae became active in the subsequent frame (*Figure 2G*). In contrast, following 6 s of inactivity, only 19% of larvae became active spontaneously; longer bouts of quiescence were not associated with a further reduction in probability of subsequent spontaneous activity (*Figure 2G*). These results suggest that ≥6 s of quiescence represents an altered behavioral state in comparison to brief (2 s) periods of inactivity. To test directly whether arousal threshold is altered with ≥6 s of behavioral quiescence, we presented a 1 s high intensity blue light stimulus and assessed arousal in larvae that were inactive at the time of stimulus presentation. While 82.4% of larvae were aroused after 2 s of inactivity, only 46.5% were aroused by the same stimulus if quiescent for ≥6 s prior to light presentation (n = 108 quiescent episodes for 2 s and 357 quiescent episodes for ≥6 s; p<0.0001, Fisher's exact test). Together, these findings indicate that larval quiescence is a reversible state during which arousal threshold is increased from baseline, and that behavioral quiescence lasting at least 6 s can be defined as sleep.

Does larval quiescence exhibit homeostatic properties of sleep, with rebound following enforced sleep loss? We perturbed sleep using blue light pulses over a 3 hr window, which resulted in 40% reduction in sleep during that time (*Figure 3A–C*). Over the subsequent 3 hr, sleep deprived larvae exhibited increased quiescence compared to non-deprived controls (*Figure 3C*), resulting primarily from increased quiescence bout length (*Figure 3D,E*). Use of a mechanical stimulus to disturb quiescence likewise resulted in subsequent quiescence rebound (*Figure 3—figure supplement 1*), showing that this rebound is not specific to the modality of quiescence loss. During rebound, sleep-deprived larvae also exhibit deeper sleep, as evidenced by an increased arousal threshold compared to non-deprived larvae (*Figure 3—figure supplement 2*). Importantly, disrupting quiescence did not cause long-lasting deficits to larval rest/activity or disruption of normal developmental progression

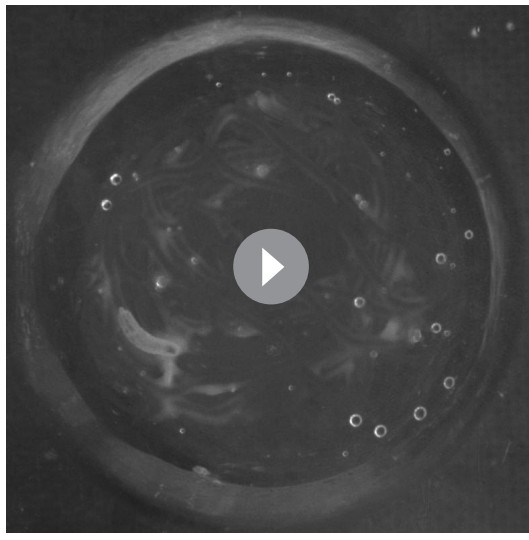

**Video 4.** Time-lapse (36x) of a second instar larva exhibiting a postural change (head retraction) associated with quiescence. Quiescence bout begins at 00:09, followed by postural change at 00:11, and then continued quiescence.
DOI: https://doi.org/10.7554/eLife.33220.009

(*Figure 3—figure supplement 1*), demonstrating that increased quiescence immediately following deprivation does not reflect a nonspecific impairment or injury related to deprivation.

In most animals, including adult fruit flies, sleep timing is controlled by a circadian clock (*Borbély, 1982*). We examined the role of circadian factors in *Drosophila* larval sleep. Anatomical analysis has shown that clock cells are present in *Drosophila* larvae (*Liu et al., 2015*), and light exposure in first instars results in synchronized adult behavioral rhythms, indicating that photic cues for circadian entrainment can be perceived at this stage (*Sehgal et al., 1992*). However, we found that rearing animals in constant light, which disrupts the circadian clock, did not change features of larval sleep (*Figure 4A*; *Figure 4—figure supplement 1A, B*), including the propensity for increased sleep in the early second instar stage. Behavioral analysis of the circadian clock mutants *clock^{jrk}* and *cyc^{01}* likewise revealed no alteration of larval sleep (*Figure 4B*; *Figure 4—figure supplement 1C,D*), suggesting that circadian factors do not play a prominent role in developmentally timed sleep in *Drosophila* larvae.

We sought to use *Drosophila* larvae to understand mechanisms of sleep control during development, and first examined whether previously characterized adult short-sleeping mutants exhibit sleep deficits as larvae. Surprisingly, mutants for the gene *sleepless* (*Koh et al., 2008*), which regulates potassium channel activity, and for the gene *fumin* (*Kume et al., 2005*), which encodes a dopamine transporter, both showed normal larval sleep levels during the second instar period (*Figure 4C*; *Figure 4—figure supplement 2*), despite exhibiting severe short-sleeping phenotypes in adulthood. We next asked whether sleep as larvae is predictive of sleep in adulthood. Sleep amount was assessed in second instar larvae, which were subsequently collected and housed individually into adulthood, at which time sleep was again measured, revealing that sleep of individual larvae was not correlated with their sleep as adults (*Figure 4D*). Finally, in contrast to adulthood when female flies sleep less than males (*Andretic and Shaw, 2005*), we did not observe sexual dimorphisms to larval sleep (*Figure 4D*). Taken together, these data indicate that genetic controls of developmental and adult sleep are at least partially distinct.

## Octopamine modulates sleep-wake in larvae

To gain insight into the cellular control of larval sleep, we next examined whether the neurotransmitter systems dopamine (DA) and octopamine (OA), which are known to be wake-promoting in adult flies (*Andretic et al., 2005*; *Crocker et al., 2010*; *Kume et al., 2005*), function analogously in larvae. Both DA and OA

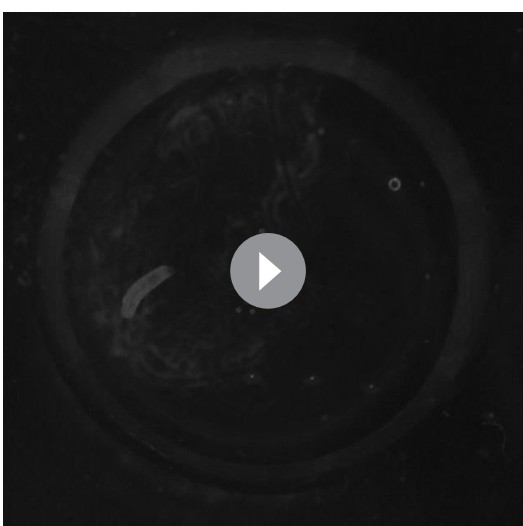

**Video 5.** Time-lapse (36x) of a second instar larva demonstrating rapid reversibility of quiescence. Quiescence begins at 00:02, and the light stimulus occurs at 00:05.
DOI: https://doi.org/10.7554/eLife.33220.010

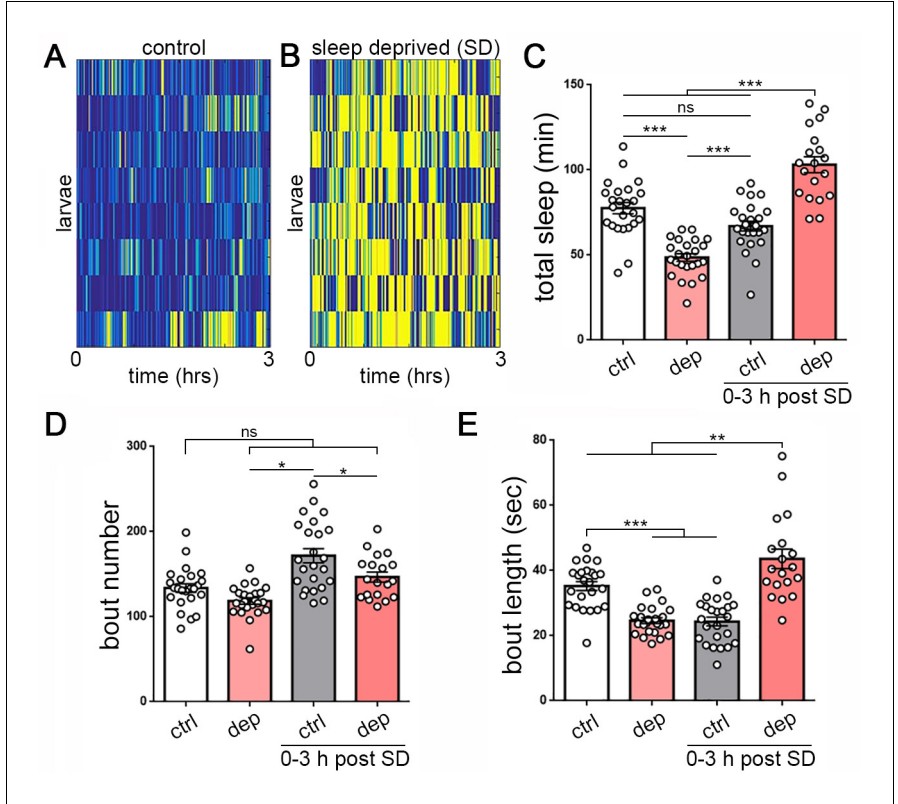

**Figure 3.** Homeostatic sleep rebound following enforced sleep loss in *Drosophila* larvae. (A, B) Activity heat map of 8 control larvae (A) and eight sleep-deprived larvae (B) during light-based sleep deprivation assay demonstrates reduced quiescence (blue) and increased activity (yellow) throughout the 3 hr deprivation period. (C) Quantification of sleep deprivation (SD) over 3 hr with a repetitive light stimulus (white = control, n = 24 larvae; light red = deprived, n = 24) and subsequent sleep rebound (dark red = previously deprived, n = 18; gray = non deprived assayed during the same period, n = 24). (D,E) Quantification of sleep bout number and length during the 3 hr deprivation period and subsequent rebound (0–3 hr post SD) demonstrates that sleep loss derives from reduced sleep bout length (E). Increased total sleep during the rebound period is more consolidated, with reduced bout number (D) and increased bout length (E).

DOI: https://doi.org/10.7554/eLife.33220.011

The following figure supplements are available for figure 3:

**Figure supplement 1.** Homeostatic sleep rebound following mechanical sleep deprivation (SD).
DOI: https://doi.org/10.7554/eLife.33220.012

**Figure supplement 2.** Increased sleep depth following sleep deprivation.
DOI: https://doi.org/10.7554/eLife.33220.013

neurons are present in second instar larvae (*Figure 5A,B*). We experimentally activated larval DA or OA neurons by expressing a bacterial sodium channel (UAS-*NaChBac* [*Nitabach et al., 2006*]) under control of *TH*-GAL4 (DA) or *Tdc2*-GAL4 (OA), respectively. Excitation of OA neurons in larvae resulted in >50% reduction in sleep time (*Figure 5C*), due primarily to a reduction in sleep bout length (*Figure 5—figure supplement 1*). In contrast, excitation of DA neurons had no significant effect on larval sleep (*Figure 5C*; *Figure 5—figure supplement 1*), consistent with the absence of a sleep phenotype in *fumin* larvae (*Figure 4C*), which harbor a mutation in the dopamine transporter (*Kume et al., 2005*).

To ask whether the relevant wake-promoting neurotransmitter used by OA neurons is indeed octopamine, we conducted sleep analysis in a series of OA synthesis mutants. Mutation of tyramine β-hydroxylase (*Tβh*), which converts tyramine to octopamine, resulted in increased sleep (*Figure 5D*; *Figure 5—figure supplement 2A–C*); this phenotype could result from reduced octopamine or from increased tyramine, which builds up in the absence of TβH. To distinguish between these possibilities, we examined a tyrosine decarboxylase (*Tdc2*) mutant that fails to convert tyrosine to tyramine,

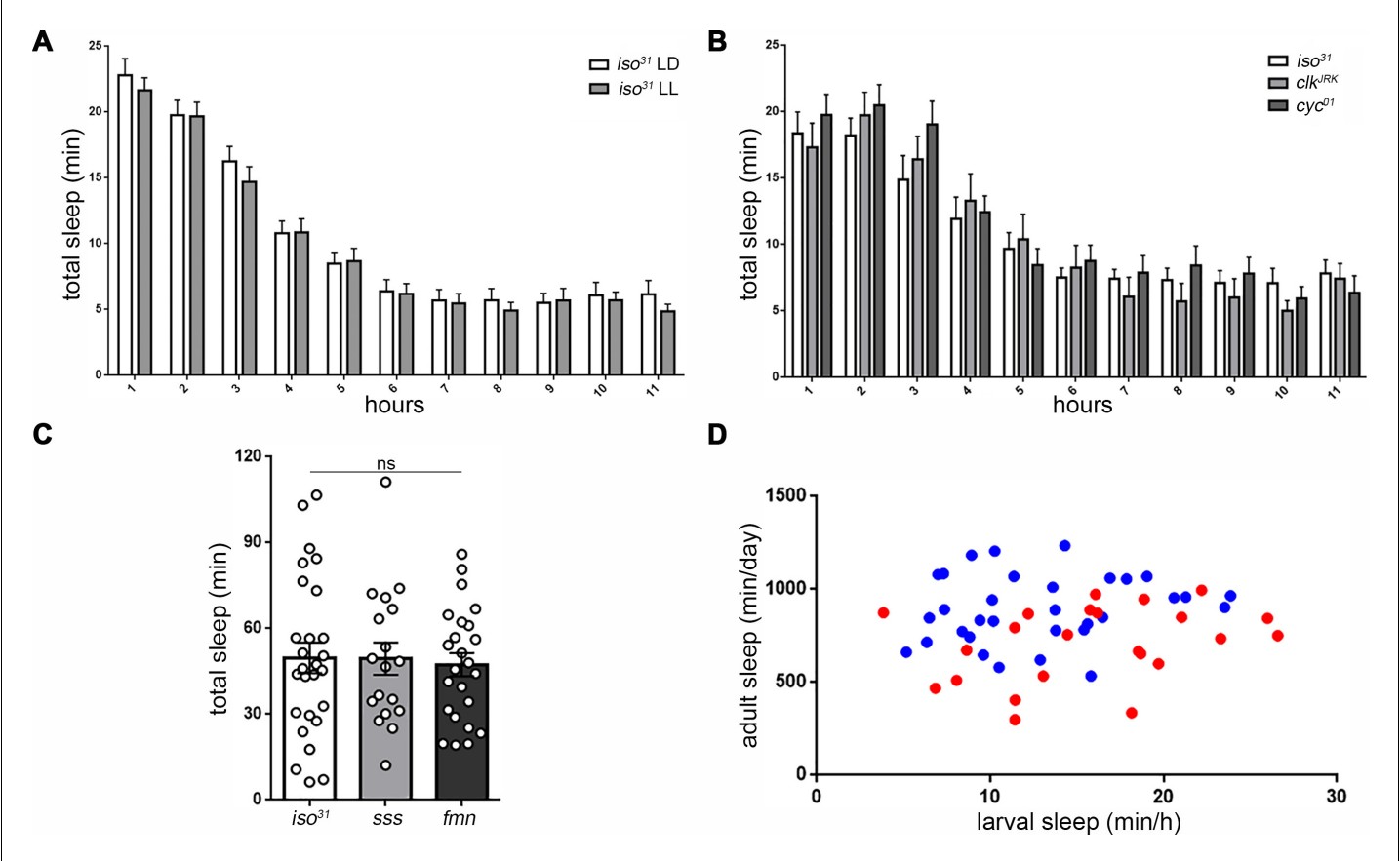

**Figure 4.** Sleep regulatory mechanisms are distinct between larval and adult stages in *Drosophila*. (A) Quantification of sleep in hourly bins demonstrates that sleep amount and distribution are unchanged with rearing of embryos/larvae in constant light (LL; n = 27) compared to normal 12:12 light:dark cycles (LD; n = 29). (B) Sleep is unaffected in molecular clock mutants *clk$^{Jrk}$* and *cyc$^{01}$* (n = 16,18,21). All sleep assays were conducted in constant dark under infrared light condition. (C) Quantification of larval total sleep over 6 hr in mutants known to be short-sleepers as adults (*iso$^{31}$* controls, *sleepless* [*sss*], and *fumin* [*fmn*]; n = 27,18,24 larvae from left to right). (D) No correlation is found between larval and adult sleep time (slope of linear regression line is not significantly different from zero). Sleep is sexually dimorphic in adults (Female (F) < Male (M); the red points (F) are distributed on the y axis lower than the blue points (M)), but not in larvae (red and blue points are evenly distributed on the x axis) (M; n = 34; F; n = 25).

DOI: https://doi.org/10.7554/eLife.33220.014

The following figure supplements are available for figure 4:

**Figure supplement 1.** Temporal distribution of larval sleep is independent of the circadian clock.

DOI: https://doi.org/10.7554/eLife.33220.015

**Figure supplement 2.** *Drosophila* larval sleep in adult sleep mutants.

DOI: https://doi.org/10.7554/eLife.33220.016

and therefore has low levels of both tyramine and octopamine. This mutant likewise exhibited increased sleep (*Figure 5E*; *Figure 5—figure supplement 2D–F*), indicating that reduced octopamine levels are causative. While both *Tβh* and *Tdc2* mutants show increased sleep compared to controls, only the *Tβh* mutation resulted in reduced activity, demonstrating that quiescence and activity levels are dissociable (*Figure 5—figure supplement 2C,F*). Next, to determine whether OA is necessary specifically in the larval nervous system for regulation of arousal, we expressed *Tβh* RNAi under control of the pan-neuronal driver *Elav*-GAL4. Knockdown of TβH resulted in increased sleep (*Figure 5F*; *Figure 5—figure supplement 3*), indicating that octopamine functions in larval neurons to control arousal. Lastly, the OA receptor subtype OAMB has been implicated in octopamine-dependent wake in adult flies (*Crocker et al., 2010*). We examined larval sleep in an *Oamb* null mutant, and found that these larvae exhibit increased sleep (*Figure 5G*; *Figure 5—figure supplement 4*). In contrast, mutation of a distinct OA receptor subtype Octβ2R did not affect sleep

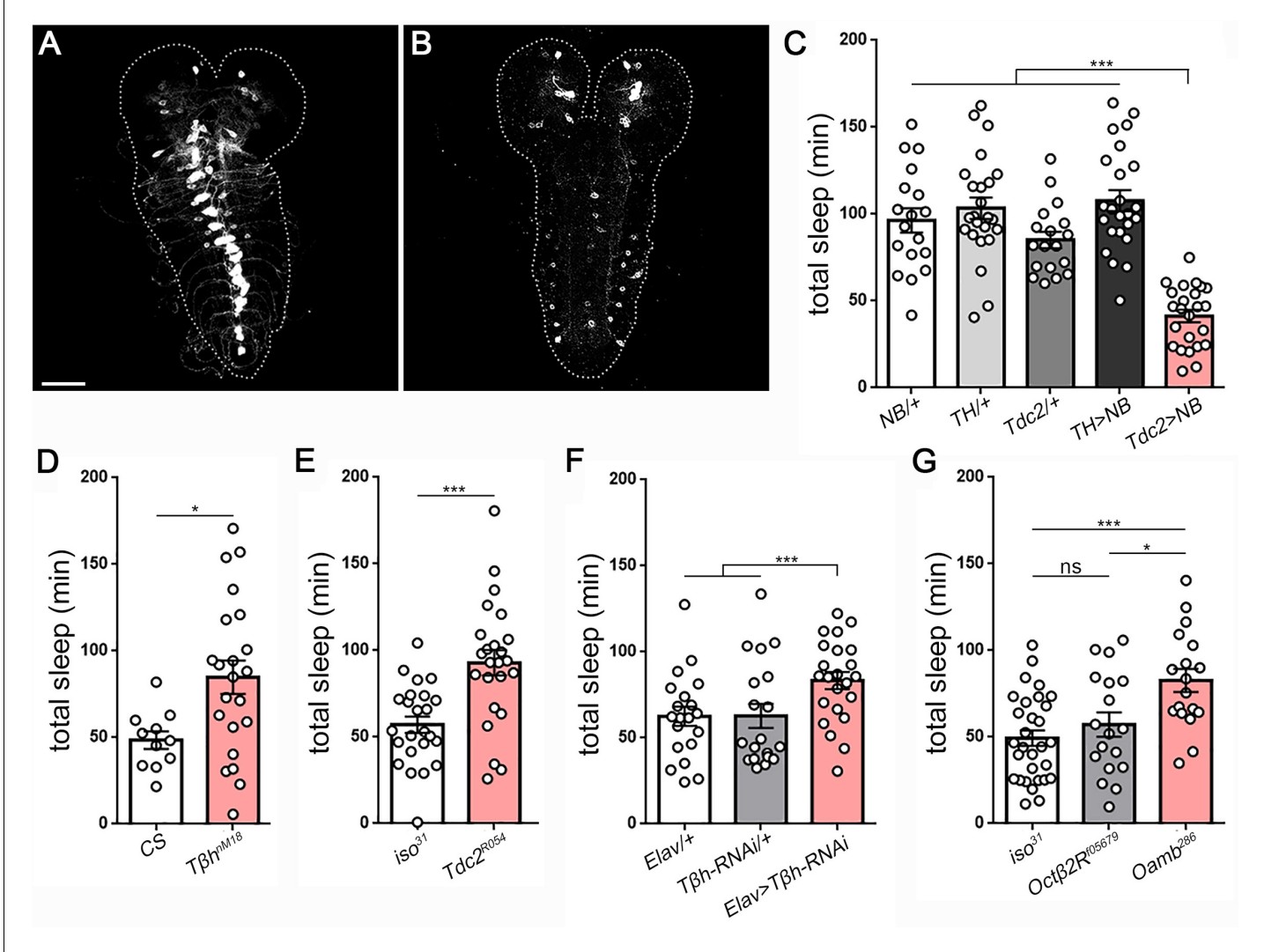

**Figure 5.** Octopamine controls sleep/wake in *Drosophila* larvae. Second instar larval brain and ventral nerve cord showing GFP expression in (**A**) octopamine neurons (*Tdc2*-GAL4 > UAS-*CD8::GFP*) and (**B**) dopamine neurons (*TH*-GAL4 >UAS-*CD8::GFP*). Scale bar = 50 μm. (**C**) Total sleep with activation of octopamine neurons (red bar; *Tdc2*-GAL4 > UAS *NachBac* [*Tdc2 >NB*]) or dopamine neurons (*TH*-GAL4 >UAS *NachBac* [*TH >NB*]), and genetic controls (n = 18,24,18,24,24). Larval sleep in *Tβh* mutants (**D**) (n = 11,21), *Tdc2* mutants (**E**) (n = 24,24), following Tβh knockdown in the nervous system (*Elav*-GAL4 >UAS-*Tβh* RNAi) (**F**) (n = 20,21,23), and in octopamine receptor mutants (**G**) (n = 30,18,19).

DOI: https://doi.org/10.7554/eLife.33220.017

The following figure supplements are available for figure 5:

**Figure supplement 1.** Activation of octopamine neurons reduces sleep in *Drosophila* larvae.

DOI: https://doi.org/10.7554/eLife.33220.018

**Figure supplement 2.** Measures of larval sleep quality in octopamine synthesis mutants.

DOI: https://doi.org/10.7554/eLife.33220.019

**Figure supplement 3.** Measures of larval sleep quality following knockdown of *Tβh* in the nervous system.

DOI: https://doi.org/10.7554/eLife.33220.020

**Figure supplement 4.** Measures of larval sleep quality in OAMB mutants.

DOI: https://doi.org/10.7554/eLife.33220.021

amount (*Figure 5G*). Together, these results identify OA as a key arousal-promoting signal during larval development.

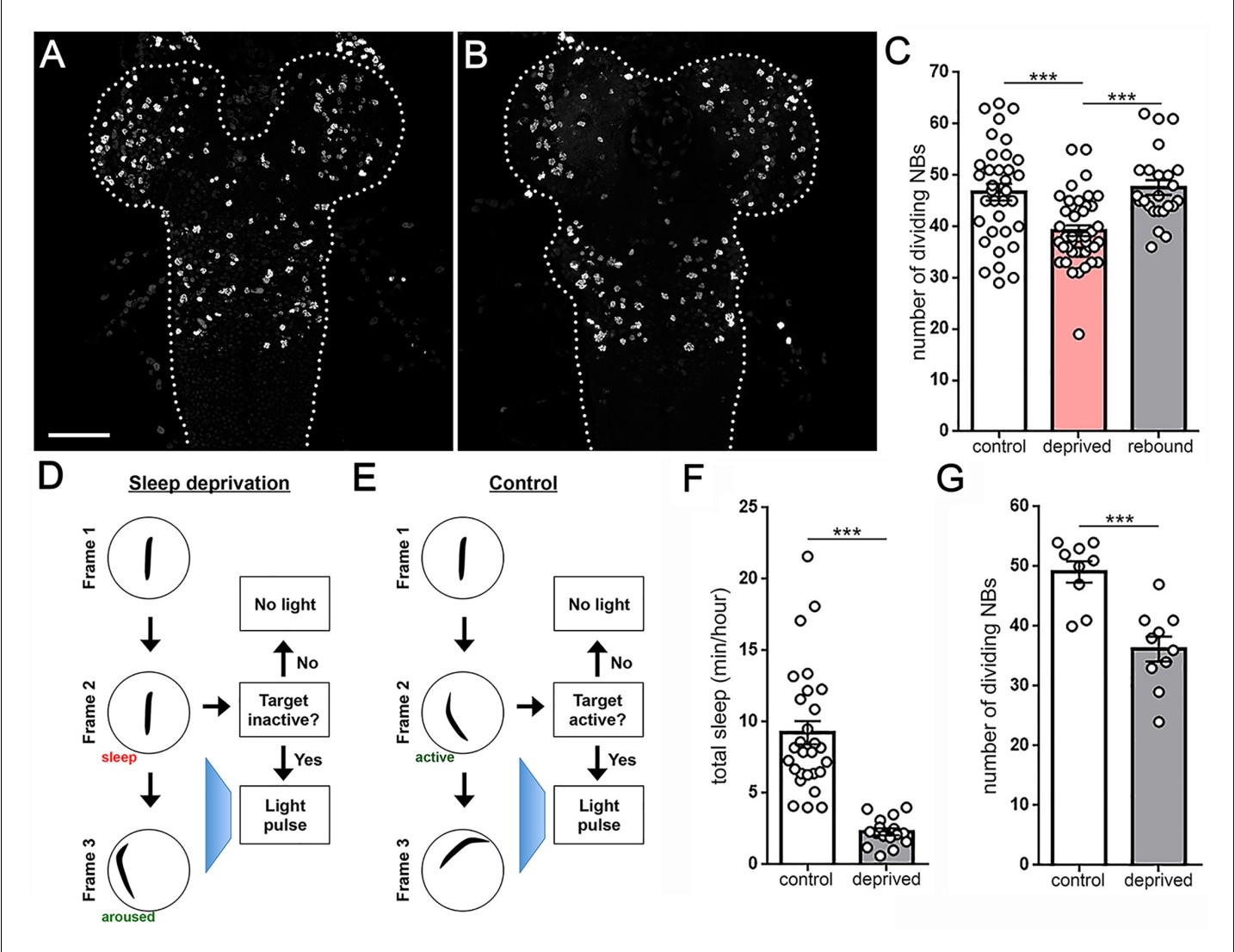

**Figure 6.** Larval sleep deprivation attenuates proliferation of neural progenitor cells. Image of second instar larval brain and ventral nerve cord (dashed outline) labeled with anti-PH3 (dividing cells) following normal sleep (**A**) or 3 hr of sleep deprivation (**B**). (**C**) Quantification of dividing cells in controls, after sleep deprivation, and following recovery sleep (n = 35, 42, 24 larvae per condition). (**D,E**) Schematic of closed-loop sleep deprivation system (**D**) and control (**E**). (**F**) Total sleep per hour in larvae exposed to light stimulus in a closed-loop system during only wake (control, n = 28) or only sleep (deprived, n = 16). (**G**) Quantification of dividing cells in controls (n = 9) or after sleep deprivation (n = 10) using a closed-loop system. Scale bar = 40 μm.

DOI: https://doi.org/10.7554/eLife.33220.022

The following figure supplement is available for figure 6:

**Figure supplement 1.** Total neuroblast number is not altered by sleep deprivation.

DOI: https://doi.org/10.7554/eLife.33220.023

## Sleep deprivation attenuates proliferation of neural progenitor cells

The ability to manipulate sleep in larvae opens new avenues to study the function of sleep in development. Type I neuroblasts (NBs) make up the majority of stem cells in the larval nervous system; these NBs undergo asymmetric divisions to self-renew and form a ganglion mother cell (GMC), which then produces two differentiated cells (neurons/glia) (*Homem and Knoblich, 2012*; *Kohwi and Doe, 2013*). The first wave of neurogenesis in embryos produces only 10% of neurons in the adult brain; onset of the second wave of neurogenesis, during which 90% of neurons in the adult brain are born, largely coincides with the second instar period (*Homem and Knoblich, 2012*). We tested whether

sleep in second instar larvae regulates nervous system stem cell proliferation. Approximately 2 hr following the molt to second instars, larvae were sleep deprived for 3 hr using a mechanical stimulus. Immediately after sleep deprivation, we assessed the number of dividing NBs in the ventral nerve cord using the mitotic marker phospho-histone H3 (PH3) in conjunction with the neural stem cell marker Miranda. Strikingly, we observed a ~15% reduction in the number of dividing NBs following mechanical sleep deprivation in comparison to non-deprived controls (*Figure 6A–C*). We detected no difference in total NB number (dividing and non-dividing) between deprived (156.1 ± 1.6 NBs) or control groups (155.7 ± 2.3 NBs) (*Figure 6—figure supplement 1*), confirming that mechanical stimulation selectively impairs dividing cells without affecting the total reservoir of NBs. To determine whether this effect on proliferation is reversible, we examined cell division in the nervous system of larvae that were sleep deprived and then permitted to sleep rebound for the subsequent hour. Following recovery sleep, the number of dividing cells returned to baseline (*Figure 6C*), demonstrating that suppression of neuronal proliferation with sleep deprivation is not a permanent impairment.

Is sleep loss the key component to reduced rate of cell division? To answer this question, we developed a closed-loop sleep deprivation system to test for a specific effect of sleep loss on neural stem cell proliferation. In this system, an individual larva was tracked in real time and illuminated with a 4 s pulse of blue light whenever a 6 s period of quiescence was detected (*Figure 6D*). As a control, the system was reversed to deliver the same total amount of light stimulus, but only during periods of activity (*Figure 6E*). Sleep/activity for all larvae was monitored throughout the experiment. This approach yielded a dramatic reduction in sleep of target larvae compared to controls (*Figure 6F*), despite all larvae receiving the same total stimulus (53.0 ± 0.99 light pulses, control; 50.6 ± 5.19 light pulses, deprived), demonstrating the ability to use this system to rule out a non-specific stimulus-dependent effect on cell division rate. Consistent with results following mechanical sleep loss, sleep deprivation using this closed-loop light-based system caused a reduction in the number of dividing NBs (*Figure 6G*). In contrast, larvae receiving the same total stimulus during active periods and without sleep loss showed no change in cell division (*Figure 6G*). Collectively, our data demonstrate that sleep loss during development is associated with reduced proliferation of neural progenitor cells.

## Discussion

A critical function for sleep in the developing brain has been hypothesized for more than 50 years (*Roffwarg et al., 1966*), supported by evidence from numerous animal models. Most of these studies have focused on early postnatal life (in mammals) or equivalent periods of elevated brain plasticity in other systems (such as young adult fruit flies) (*Blumberg et al., 2013*; *Cirelli and Tononi, 2015*; *Frank, 2011*; *Kayser et al., 2014*). However, investigation of a role for sleep during even earlier developmental time points has been limited by experimental inaccessibility of mammals to *in utero* sleep monitoring and deprivation. Here, we develop a platform for long-term monitoring of behavior in *Drosophila* larvae. We demonstrate that *Drosophila* larvae exhibit periods of reversible behavioral quiescence characterized by a change in posture and increased arousal threshold, with homeostatic properties such that quiescence deprivation is followed by subsequent increased quiescence amount and depth. These features meet the behavioral criteria for sleep shared across phylogeny, and establish *Drosophila* larvae as a new system to study questions at the intersection of sleep and development.

Little is known regarding the cellular and molecular control of sleep during early life. Surprisingly, we find that the most severe short sleeping mutations identified in adult flies exhibit no sleep phenotype in larvae, and that sleep time in larvae is not correlated with that of adults. Moreover, direct activation of dopamine neurons in second instar larvae fails to promote wakefulness as it does in adult flies. Together these findings suggest that genetic and neural sleep regulatory mechanisms might be partially distinct at different developmental time points. What purpose might be served by this mechanistic divergence? One possibility is that the drives facing a developing animal (e.g., high levels of feeding to meet metabolic demands related to tissue growth) are different than those in adulthood (e.g., reproductive drive). Sleep must be suppressed to fulfill these demands (*Beckwith et al., 2017*; *Chen et al., 2017*; *Keene et al., 2010*; *Machado et al., 2017*), and perhaps sleep regulatory mechanisms are dynamically coupled to the most relevant environmental cues for drives that change throughout maturation. Notably, sleep bouts in *Drosophila* larvae are relatively

brief, with an average length of 15–20 s. We identify 6 s as the minimum period of behavioral quiescence considered sleep. Though brief, sleep bouts of this length are similar to that described in other organisms, including adult zebrafish in which inactivity for 6 s is also the minimum period defined as sleep (*Yokogawa et al., 2007*). Moreover, the fragmented nature of sleep we observe in larvae is consistent with the idea that rapid sleep-wake cycling is a shared feature of early life sleep across phylogeny: human infants exhibit fragmented sleep (*Kleitman and Engelmann, 1953*), and early postnatal rats show sleep bout durations averaging <15 s that begin to consolidate in the first postnatal weeks (*Blumberg et al., 2005a*). Fragmented sleep in *Drosophila* larvae (and perhaps other young animals) might reflect high metabolic needs of a rapidly developing organism, raising the possibility that sleep regulatory mechanisms during development show divergence from adulthood in order to facilitate brief rest bouts interspersed among frequent feeding required to meet the developmental challenge.

Not all sleep-wake systems are distinct between *Drosophila* larvae and adults, as octopamine is a key arousal-promoting signal in both. Activation of OA neurons in larvae promotes wake while OA synthesis or receptor mutants exhibit more sleep. Given the conserved role of OA/NE in mammalian sleep-wake (*Aston-Jones and Bloom, 1981*), this neurotransmitter may also be a critical regulator of early developmental sleep in other species including humans. Our results are consistent with the idea that sleep control from early life to maturity is characterized by sequential elaboration of regulatory mechanisms (*Blumberg et al., 2005b*; *Cirelli and Tononi, 2015*; *Karlsson et al., 2005*), perhaps starting with noradrenergic signaling. In the adult fly, specific clusters of OA-releasing neurons mediate arousal/wake (*Crocker et al., 2010*). Whether such circuit-specific features exist in larval sleep is not known. While OA serves an arousal function, we have yet to identify a sleep-promoting brain region in larvae. Sleep/arousal centers are organized with common neural logic across phylogeny, where activity of one region dampens the other to switch between states (*Artiushin and Sehgal, 2017*; *Saper et al., 2005*). Specifically, in the adult fly brain, wake-promoting neurons project to and inhibit activity of a sleep-promoting area (*Liu et al., 2012*; *Ueno et al., 2012*). A circuit-level understanding of sleep/wake control during development is necessary for studying sleep ontogeny across species. Given the detailed understanding of sleep circuitry in adult flies (*Artiushin and Sehgal, 2017*), the relatively simple larval nervous system presents a unique opportunity to examine how sleep-arousal circuitry in early development compares to adulthood.

Most physiological functions, including cell cycle progression and DNA replication, are gated within optimal time windows (*Matsuo et al., 2003*; *Tu et al., 2005*). Work in adult rodents suggests that sleep restriction/fragmentation reduces hippocampal neurogenesis (*Guzmán-Marín et al., 2003*; *Lucassen et al., 2010*), but it has been difficult to untangle the effect of sleep loss per se from the well-known impairment of neurogenesis with stress of sensory stimuli used to keep animals awake (*Lucassen et al., 2010*; *Warner-Schmidt and Duman, 2006*); sleep deprivation paradigms have, for the most part, not controlled for this issue (*Lucassen et al., 2010*). Moreover, the magnitude of neurogenesis (and therefore consequences of its disruption) is dramatically higher in early development compared to adulthood. The timing of a major wave of neurogenesis to larval life, a period accessible for experimental manipulation, makes *Drosophila* larvae ideal for examining sleep function and control during early development. Using a closed-loop system in which control animals received the same degree of sensory stimulation as sleep-deprived larva, we uncovered a previously unknown role for sleep in developmental neurogenesis, with broad implications for both brain patterning and the ramifications of sleep loss in early life.

A number of areas for investigation emerge from this work. First, while sleep quantity is greater in early second instar larva compared to later (*Figure 1D*), the rate of neuronal proliferation does not decrease over this same period (*Ito and Hotta, 1992*). Together, these findings suggest sleep might act as a permissive, but not instructive, cue for neurogenesis. Alternatively, sleep could play an important role in initiating the wave of neurogenesis, though not be essential to sustain this process. Second, we provide evidence that sleep can influence NB proliferation, but have not established whether this relationship is bidirectional. While critical to control for more general effects on larval health and function, it will be interesting to dissect how accelerating/slowing cell division affects sleep. Finally, it remains unknown whether the effect of sleep loss on neuronal proliferation is general, or if specific NB subtypes are particularly sensitive to sleep deprivation, which could lead to predictable deficits following sleep loss. Our work describes an ~15% reduction in rate of cell division with 2–3 hr of sleep loss. We aim to develop approaches that achieve long-lasting sleep

deprivation, with the goal of understanding how impaired neurogenesis during early development affects brain and behavioral development. More broadly, future work will leverage the tractable larval system to gain novel insights into the genetic/molecular regulation of sleep during development and examine how sleep and neurogenesis are mechanistically coupled.

# Materials and methods

## Key resources table

| Reagent type (species) or resource | Designation | Source or reference | Identifiers | Additional information |
|---|---|---|---|---|
| gene (Drosophila melanogaster) | clock | NA | FLYB:FBgn0023076 | |
| gene (D. melanogaster) | cycle | NA | FLYB:FBgn0023094 | |
| gene (D. melanogaster) | sleepless | NA | FLYB:FBgn0260499 | |
| gene (D. melanogaster) | DAT | NA | FLYB:FBgn0034136 | |
| gene (D. melanogaster) | Oamb | NA | FBgn0024944 | |
| gene (D. melanogaster) | Octβ2R | NA | FBgn0038063 | |
| gene (D. melanogaster) | Tdc2 | NA | FBgn0050446 | |
| gene (D. melanogaster) | Tbh | NA | FBgn0010329 | |
| genetic reagent (D. melanogaster) | iso31 | Dr. Amita Sehgal | | |
| genetic reagent (D. melanogaster) | clockjrk | *Allada et al., 1998* | FLYB:FBal0090722 | |
| genetic reagent (D. melanogaster) | cyc01 | *Rutila et al., 1998* | FBal0195440 | |
| genetic reagent (D. melanogaster) | sssP1 | *Koh et al., 2008* | FBal0121566 | |
| genetic reagent (D. melanogaster) | fumin | *Kume et al., 2005* | FBal0197506 | |
| genetic reagent (D. melanogaster) | Oamb286 | *Lee et al., 2003* | FBal0152344 | |
| genetic reagent (D. melanogaster) | Octβ2Rf05679 | Bloomington Drosophila Stock Center | FBal0161089 | |
| genetic reagent (D. melanogaster) | Tdc2R054 | *Cole et al., 2005* | | |
| genetic reagent (D. melanogaster) | TbhnM18 | *Monastirioti et al., 1996* | FBal0061578 | |
| genetic reagent (D. melanogaster) | UAS-NachBac | *Nitabach et al., 2006* | FBtp0021523; BDSC:9469 | |
| genetic reagent (D. melanogaster) | TH-Gal4 | *Friggi-Grelin et al., 2003* | FBtp0114847; BDSC:8848 | |
| genetic reagent (D. melanogaster) | Tdc2-Gal4 | Bloomington Drosophila Stock Center | FBtp0056985; BDSC:9313 | |
| genetic reagent (D. melanogaster) | Elav-Gal4 | Bloomington Drosophila Stock Center | FBtp0000743; BDSC:8765 | |
| genetic reagent (D. melanogaster) | UAS-CD8::GFP | Bloomington Drosophila Stock Center | FBtp0002652 | |
| genetic reagent (D. melanogaster) | Tbh-RNAi | Vienna Drosophila Resource Center | FBst0478893; VDRC:107070 | |
| antibody | anti-Miranda | Abcam | Abcam:ab197788 | (1:50) |
| antibody | anti-Ph3 | Invitrogen | Invitrogen: PA5-17869 | (1:1000) |
| antibody | anti-GFP | ThermoFisher Scientific | ThermoFisher Scientific: A-11122 | (1:1000) |

## *Drosophila* stocks

The following lines have been maintained as lab stocks or were obtained from Dr. Amita Sehgal: $iso^{31}$, $clock^{jrk}$ (*Allada et al., 1998*), $cyc^{01}$ (*Rutila et al., 1998*), $sss^{P1}$ (*Koh et al., 2008*), $fmn$ (*Kume et al., 2005*), UAS-*NachBac* (*Nitabach et al., 2006*), TH-Gal4 (*Friggi-Grelin et al., 2003*), $Oamb^{286}$ (*Lee et al., 2003*), $Tdc2^{R054}$ (*Cole et al., 2005*), $Tbh^{nM18}$ (*Monastirioti et al., 1996*) and CS control. Tdc2-Gal4, Elav-Gal4, UAS-*CD8::GFP*, and $Octβ2R^{f05679}$ are from the Bloomington Stock Center. *Tbh*-RNAi was obtained from the Vienna Drosophila Resource Center (VDRC#: 107070).

## Larval rearing and sleep assays

Adult flies were maintained on standard molasses-based diet at 25°C on a 12:12 light:dark (LD) cycle. In order to collect synchronized second instar larvae, adult flies were placed in an embryo collection cage (Genesee Scientific, cat#: 59–100) and eggs were laid on a petri dish containing 3% agar, 2% sucrose, and 2.5% apple juice with yeast paste on top. Animals developed on this media for two days. On the day of the sleep assay, molting first instar larvae were collected and moved to

a separate dish with yeast to complete the molt to second instar. Freshly molted second instars were carefully placed into individual wells of the LarvaLodge containing 100 µl of 3% agar and 2% sucrose media covered with a thin layer of yeast paste. The LarvaLodge was covered with a transparent acrylic sheet and placed into a DigiTherm (Tritech Research) incubator at 25°C for imaging. Unless otherwise indicated, experiments were performed in constant dark.

## LarvaLodge design, fabrication, and preparation

LarvaLodge design was adapted from the WorMotel, a custom microplate consisting of individual wells developed for monitoring of *C. elegans* (*Churgin et al., 2017*). We designed a chip containing a rectangular array of rounded wells with 11 mm diameter and 3 mm depth. Designs of the LarvaLodge masters were created using MATLAB. We printed a master corresponding to the negative of this shape with an Objet30 photopolymer 3D printer using the material VeroBlack. To mold the devices, we mixed Dow Corning Sylgard 184 PDMS according to the manufacturer's instructions and poured 20 g into the master. Devices were cured overnight at room temperature and then removed from molds using a spatula.

## Image acquisition

Images were captured every 6 s with an Imaging Source DMK 23GP031 camera (2592 × 1944 pixels, The Imaging Source, USA) equipped with a Fujinon lens (HF12.5SA-1, 1:1.4/12.5 mm, Fujifilm Corp., Japan) with a Hoya 49mm R72 Infrared Filter. We used IC Capture (The Imaging Source) to acquire time-lapse images through a gigabit Ethernet connection. All experiments were carried out under dark-field illumination using infrared LED strips (Ledlightsworld LTD, 850 nm wavelength) positioned below the LarvaLodge.

## Image processing

Images were analyzed using custom-written MATLAB software. Temporally adjacent images were subtracted to generate maps of pixel value intensity change (*Churgin et al., 2017*; *Raizen et al., 2008*). A binary threshold was applied such that individual pixel intensity changes that fell below 50 gray-scale units within each region of interest (individual lodge) were set equal to zero ('not changed') to eliminate noise, whereas pixel changes greater than or equal to 50 gray-scale units were set equal to one ('changed'). We then calculated the activity, defined as the sum of all pixels changed between images (i.e., the total number of pixels where a change in intensity occurred above the noise threshold). Quiescence was defined as an activity value of zero between frames. Total quiescence (sleep) was summed in hourly bins (*Figures 1* and *4A,B*) or over 6 hr beginning 2 hr after the molt to second instar (*Figures 4C* and *5*). Timing of sleep deprivation experiments (*Figures 3* and *6*) is provided below.

## Feeding behavior analysis

Freshly molted wild type second instar larvae were starved on 3% agar plates for 2 hr and placed into LarvaLodges with agar/sucrose covered by yeast paste containing red food coloring. Activity was captured for 5 min (6 s per frame). Larvae were removed and inspected for red color accumulated in their guts to verify feeding during the monitoring period.

## Postural change analysis

107 long sleep bouts (36 s or more) and 104 shorter sleep bouts (18 s) from six animals were selected for further analysis. Each sleep bout was manually analyzed, including three frames (18 s) before and after the period of inactivity. We noticed that postural changes most commonly bridged two otherwise continuous period of inactivity, and less often occurred at the initiation or termination of quiescence. Head retraction and body proportion measurement (NIH ImageJ) were made on all long sleep bouts containing a postural change (n = 82), measuring the animals from the tip of the head to the end of the tail, and across the body half way along the anterior-posterior axis. Measurements were made before and after the postural change.

## Arousal and sleep deprivation with blue light stimulation

To supply the blue light illumination, we used two high power LEDs (Luminus Phatlight PT-121, 460 nm peak wavelength, Sunnyvale, CA) secured to an aluminum heat sink. We used a relay (Schneider Electric, France) controlled by MATLAB through a LabJack (LabJack Corp., Lakewood, CO) to drive the LEDs at a current of either 0.1 A (low intensity, 3.98 μW/mm$^2$) or 1 A (high intensity, 39.8 μW/mm$^2$) for 4 s through a power supply. We measured blue light irradiance using a power meter (LaserMate-Q, Coherent, Santa Clara, CA). To maximize the blue light uniformity delivered to the microplate, we constructed a mirror box consisting of four mirrored-acrylic panels with mirrored sides facing inwards (Churgin et al., 2017) and placed it around the LarvaLodge. For light-based sleep deprivation, we used a high intensity stimulus for 30 s every 2 min for 3 hr beginning 2 hr after the molt to second instar. Undisturbed control animals were placed in a separate incubator during deprivation, and both lodges were moved to the same incubator during the rebound.

## Mechanical sleep deprivation

For mechanical sleep deprivation, LarvaLodges were placed on a Trikinetics vortexer mounting plate, with shaking for 2 s randomly within every 10 s window for 3 hr beginning 2 hr after the molt to second instar. Non-deprived larvae were housed in a separate incubator during this period. After 3 hr, half of the larvae were removed from the lodges and their brains were dissected while the other half remained in the LarvaLodges and placed into the same incubators to allow for rebound. Remaining animals were then dissected after 1 hr and processed for immunostaining.

## Closed-loop sleep deprivation system

For closed-loop sleep deprivation with blue light stimulation, the activity of individual second instar larvae was tracked in real time over 2 hr beginning 2 hr after the molt to second instar. Individual larvae were stimulated with a high intensity (39.8 μW/mm$^2$) light pulse for 4 s whenever a > 6 s period of quiescence was detected. For controls, individual animals in a separate experiment were tracked in real time and stimulated with the same light pulse for 4 s whenever a > 18 s period of continuous activity was detected. After stimulation, a refractory period of two minutes was enforced during which the control larvae would not be stimulated again. This refractory period enabled us to ensure that control (stimulated only during periods of activity) and experimental (stimulated only during periods of quiescence) animals experienced approximately the same total number of blue light stimulation pulses. Individual larva were immediately dissected at termination of the experiment and processed for immunostaining.

## Immunohistochemistry and imaging

Brains were dissected in PBS, fixed in 4% PFA for 20 min at room temperature. Following 3 × 10 min washes in PBST, brains were incubated with primary antibody at 4° overnight. Following 3 × 10 min washes in PBST, brains were incubated with secondary antibody for 2 hr at room temperature. Following 3 × 10 min washes in PBST, brains and ventral nerve cords were mounted in 90% Glycerol. Primary antibodies included: Rat anti-Miranda (1:50, ab197788, Abcam), Rabbit anti-Ph3 (1:1000, PA5-17869, Invitrogen), and Rabbit anti-GFP (1:1000, A-11122, ThermoFisher Scientific). Secondary antibodies included: FITC donkey anti-rat (1:200, Jackson), Cy5 donkey anti-rabbit (1:200, Jackson). Brains were visualized and imaged with a TCS SP5 or SP8 confocal microscope. An observer who was blinded to the experimental condition manually counted total NBs (Miranda positive) and dividing NBs (Miranda and PH3 positive) in the ventral nerve cord from 1.5 μm step confocal stacks using NIH ImageJ.

## Statistical analysis and data reproducibility

Analysis was done using Prism (GraphPad Software). ANOVA with Tukey's test was used in *Figure 1*; *Figure 2F*; *Figure 3C–E*; *Figure 4C*; *Figure 5C,F,G*; *Figure 6C*; *Figure 3—figure supplement 1*; *Figure 4—figure supplement 2*; *Figure 5—figure supplements 1* and *3*. Student's t-test was used in *Figure 2E*; *Figure 5D,E*; *Figure 6F,G*; *Figure 3—figure supplement 2*; *Figure 5—figure supplements 2* and *4*; *Figure 6—figure supplement 1*. Fisher's exact test was used in *Figure 2G*. For significance: *p≤0.05; **p<0.01; ***p<0.001. Each experiment was generated from a minimum of 3

independent biological replicates. Samples were allocated based on genotype or experimental manipulation and statistics performed on aggregated data. Outliers were never excluded.

## Acknowledgements

We thank members of the Kayser Lab and Dr. Amita Sehgal for helpful discussions. This work was supported by NIH grants K08NS090461 (MSK), R01NS088432 (DMR and CF-Y), and R01NS084835 (CF-Y), and a Burroughs Wellcome Career Award for Medical Scientists, March of Dimes Basil O'Connor Scholar Award, and Sloan Research Fellowship (MSK). MS designed and performed experiments, analyzed data, and contributed to writing the manuscript. MAC developed the Larva-Lodge, designed experiments, wrote closed-loop deprivation and analysis software, and provided technical assistance. AJB designed and performed experiments, and analyzed data. DMR designed experiments and interpreted data. CF-Y developed the LarvaLodge, provided technical assistance, and interpreted data. MSK conceived and analyzed experiments, interpreted the data, and wrote the manuscript with input from other authors.

## Additional information

### Funding

| Funder | Grant reference number | Author |
| --- | --- | --- |
| National Institutes of Health | K08NS090461 | Matthew S Kayser |
| Burroughs Wellcome Fund | | Matthew S Kayser |
| Alfred P. Sloan Foundation | | Matthew S Kayser |
| March of Dimes Foundation | | Matthew S Kayser |
| National Institutes of Health | R01NS088432 | David M Raizen |
| National Institutes of Health | R01NS084835 | Christopher Fang-Yen |

The funders had no role in study design, data collection and interpretation, or the decision to submit the work for publication.

### Author contributions

Milan Szuperak, Conceptualization, Data curation, Formal analysis, Validation, Investigation, Methodology, Writing—review and editing; Matthew A Churgin, Conceptualization, Software, Formal analysis, Investigation, Methodology, Writing—review and editing; Austin J Borja, Formal analysis, Investigation, Methodology; David M Raizen, Conceptualization, Investigation, Methodology, Writing—review and editing; Christopher Fang-Yen, Conceptualization, Software, Supervision, Investigation, Methodology, Writing—review and editing; Matthew S Kayser, Conceptualization, Formal analysis, Supervision, Funding acquisition, Validation, Investigation, Methodology, Writing—original draft, Writing—review and editing

### Author ORCIDs

Matthew A Churgin http://orcid.org/0000-0003-2299-0124
David M Raizen http://orcid.org/0000-0001-5935-0476
Christopher Fang-Yen http://orcid.org/0000-0002-4568-3218
Matthew S Kayser http://orcid.org/0000-0003-2359-4967

### Decision letter and Author response

Decision letter https://doi.org/10.7554/eLife.33220.031
Author response https://doi.org/10.7554/eLife.33220.032

## Additional files

**Supplementary files**

• Source code 1. Matlab code 'Arousal and Sleep Deprivation' was used to apply blue light stimulus with various length and interval to achieve either sleep deprivation or to produce arousal stimulus.
DOI: https://doi.org/10.7554/eLife.33220.024

• Source code 2. Matlab code 'Closed Loop Analysis' was applied to perform closed loop sleep deprivation, delivering the blue light stimulus whenever a >6 s period of quiescence was detected.
DOI: https://doi.org/10.7554/eLife.33220.025

• Source code 3. Matlab code 'Closed Loop Awake Stimulus' was used for control experiments to deliver the arousal stimulation whenever a > 18 s period of continuous activity was detected.
DOI: https://doi.org/10.7554/eLife.33220.026

• Transparent reporting form
DOI: https://doi.org/10.7554/eLife.33220.027

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
