## [Decision Letter]

Thank you for submitting your article "A sleep state in *Drosophila* larvae required for neural stem cell proliferation" for consideration by *eLife*. Your article has been reviewed by two peer reviewers, and the evaluation has been overseen by Mani Ramaswami as Reviewing Editor and Eve Marder as the Senior Editor. The following individual involved in review of your submission has agreed to reveal his identity: Mark S Blumberg (Reviewer #3).

The reviewers have discussed the reviews with one another and the Reviewing Editor has drafted this decision to help you prepare a revised submission..

Summary:

This manuscript provides a comprehensive characterization of the phenomenon of sleep in *Drosophila* larvae and demonstrates that suppressing larval sleep reduces neural proliferation. It's interesting to observe the wake-promoting effects of octopamine in larvae. Although *Drosophila* has been a well-established genetic model for sleep research for nearly two decades, this first detailed description of sleep in larvae is a substantial contribution. The finding regarding the importance of sleep for neural development is important and exciting. Together these advances open new opportunities to mechanistically study the role of sleep in neural development.

The manuscript is well written and clearly organized. However, addressing the comments below will improve the manuscript and extend the significance of the findings.

Essential revisions:

1) Sleep in second instar larvae appear highest in the few hours after molting, then tails off after several hours. The authors mention that a wave of neurogenesis also occurs in second instar larvae – are the precise kinetics of this known? If sleep plays an important role in this process, we might expect neurogenesis to be highest in the first few second instar hours during which larvae sleep peaks. This should be addressed appropriately in the Results and Discussion sections.

2) The authors show that suppressing sleep decreases neuroblast proliferation generally. Do particular neuroblasts that contribute specifically to second instar proliferation most sensitive to sleep loss? Or are neuroblasts that proliferate over wider time windows (e.g., MB neuroblasts) also affected by sleep deprivation? Is this known? And can it be addressed?

3) Neuroblast proliferation seems to be influenced by sleep, but it is not clear whether the relationship is bi-directional. Do manipulations that speed/slow proliferation also modulate sleep?

4) Is there any long-term consequence of reduced neuroblast proliferation? Or is the process just delayed by sleep deprivation? Can this be addressed or discussed?

5)The study established sleep using many established criteria: distinct sleep posture, arousal threshold, and sleep rebound after deprivation. However, it does not mention sleep pressure. A revised manuscript should provide an explanation as to why.

6) Subsection “Octopamine controls sleep-wake in larvae”: It seems too strong to state that octopamine "controls sleep-wake in larvae." Modulates? Influences?

7) Subsection “Octopamine controls sleep-wake in larvae”: It is suggested, based on a reference to the Aston-Jones and Bloom paper, that norepinephrine plays "a conserved role in developmental sleep across species." Leaving aside the awkward sentence, the reference is to an adult rat sleep paper. I don't know of any studies on norepinephrine and sleep in early development in rats or any other species, but if they exist they should be cited here. Regardless, suggestions of conserved functions at this early juncture seem premature.

8) The reviewers agree that the current analyses are perfectly adequate for the current manuscript. However, here are two possibly helpful suggestions for future manuscripts.

a) The choice of 6 seconds between video frames is longer than ideal and the authors are encouraged to reconsider this choice in the future. The average sleep bout lengths of 15-30 seconds represent only 2.5-5 frames, so 6 seconds provides a course measure of bout length; preferably, sampling intervals should be smaller in relation to bout lengths. Also, with more precise measurement of bout lengths, it will be possible to analyze the statistical distributions of the sleep and wake bout lengths, as has been done profitably in infant rats and the adults of several mammalian species. For example, do the sleep bouts distribute exponentially, as has been found in mammalian sleep? More precise measurement would also allow the authors to more sensitively assess bout consolidation, developmental changes, etc. Such analyses can also provide clues to mechanism.

b) A related issue concerns the definition of larval sleep as a minimum of 6 seconds of quiescence, which is defined as no evidence of activity across two frames. This definition, though clear, appears to be an arbitrary one. It also makes the comparison to the 6-second definition of sleep in zebrafish (Discussion section) less interesting. Why not measure sleep postures more accurately and analyze the frequency distributions of sleep times with greater precision (e.g., 1-second bins)?

---

## [Author Response]

1) Sleep in second instar larvae appear highest in the few hours after molting, then tails off after several hours. The authors mention that a wave of neurogenesis also occurs in second instar larvae – are the precise kinetics of this known? If sleep plays an important role in this process, we might expect neurogenesis to be highest in the first few second instar hours during which larvae sleep peaks. This should be addressed appropriately in the Results and Discussion sections.

We thank the reviewers for raising this interesting point. Published work (Ito and Hotta, 1992) indicates that the rate of neurogenesis does not decrease as development progresses. We find that larval sleep is highest, however, early in the second instar period. This suggests that sleep could act as a permissive, rather than instructive, cue for neurogenesis; or that sleep is important to initiate the wave of neurogenesis, but not sustain cell division. We now address these possibilities in the Discussion section.

2) The authors show that suppressing sleep decreases neuroblast proliferation generally. Do particular neuroblasts that contribute specifically to second instar proliferation most sensitive to sleep loss? Or are neuroblasts that proliferate over wider time windows (e.g., MB neuroblasts) also affected by sleep deprivation? Is this known? And can it be addressed?

We attempted to determine whether MB neuroblasts (NBs) are affected by sleep deprivation. In early first instar larvae, only MB NBs are dividing, so this population can be selectively labeled by feeding larvae EdU, which incorporates into dividing cells. We then examined second instar larval brains that were co-labeled with markers for NBs (anti-Deadpan) and dividing cells (anti-PH3). As is evident in Author response image 1, we were able to see clear MB lineages (green), though found that EdU labeling of the MB NBs themselves was already too diluted to be visible. We could still identify MB NBs (yellow arrows) in most cases based on Deadpan labeling (red) and proximity to MB lineages. However, we found that at a given point in time, of the 8 total MB NBs per brain, only 1-2 were dividing (white arrowhead; labeled by PH3 [magenta] co-localized with Deadpan [red]). Therefore, attempting to sleep deprive larvae and assess for a decrease in number of dividing MB NBs is not feasible given the low number of these NBs dividing at the time of our experiment (i.e., “floor effect”). We agree that in the future it will be interesting to label distinct subpopulations of NBs and determine whether some but not others are particularly susceptible to sleep loss. We have added discussion of this issue into the manuscript (Discussion section).

**Author response image 1. respfig1:** Labeling of MB neuroblasts (NBs) in second instar larval brains. Each panel (A-D) is a single hemisphere from four different second instar larval brains. MB lineages (green) are labeled by EdU that was pulsed for 2 hours during mid first instar stages. All NBs are labeled with anti-Deadpan (red), and dividing cells are labeled with anti-PH3 (magenta). MB NBs (yellow arrows) can be identified by location with regard to MB lineages. Most MB NBs at any point in time do not appear to be actively dividing (absence of PH3). Rarely (D, white arrow) we were able to clearly identify a MB NBs that was PH3+. Scale bar = 10μm.

3) Neuroblast proliferation seems to be influenced by sleep, but it is not clear whether the relationship is bi-directional. Do manipulations that speed/slow proliferation also modulate sleep?

In response to this question, we conducted a series of experiments in which we used genetic approaches to accelerate or slow NB cell division rate, followed by sleep assays. We used *worniu-GAL4* (Lai et al., 2012 and Lee, Robinson and Doe, 2006)to drive NB-specific expression of activated Ras (*UAS-Ras1^V12^*) or Alk (*UAS-Alk*) to increase NB proliferation (Cheng et al., 2011 and Karim and Rubin, 1998), or dp60 (*UAS-dp60*) to reduce NB proliferation (by decreasing PI3-kinase activity) (Weinkove et al., 1999 and Sipe and Siegrist, 2017). Our results were mixed, as manipulations that increased NB proliferation had different effects on sleep, while NB-specific expression of activated Ras and dp60 both increased sleep despite opposing effects on proliferation (Author response image 2). Thus, our results suggest that altering cell proliferation might indeed impinge on sleep, but that the relationship is complicated. It is likely that manipulating NB division has more general effects on the animal aside from sleep, and therefore it is difficult to draw definitive conclusions. It will be important to pursue other means of altering the rate of cell division, perhaps within more restricted NB populations, and to ensure that such manipulations do not impair overall health or function of the animal (addressed in the Discussion section).

**Author response image 2. respfig2:** Sleep following manipulation of NB division. Total sleep (**A**) and quantification of number of dividing NBs (**B**) shows that the relationship between proliferation rate and sleep is mixed. Expression of *UAS-Ras1^V12^*or *UAS-Alk* in NBs (using *worniu-GAL4*) results in a comparable increase in NB division, but only *UAS-Ras* increases sleep. In contrast, expression of *UAS-dp60* causes a dramatic reduction in dividing NBs, but sleep is increased. For sleep experiments, n=29, 41, 35, 22 larvae from left to right. For measurement of dividing NBs, n=18, 19, 14, 11 brains from left to right. ***P<0.0001. ANOVA with Tukey’s test. (We thank Dr. Sarah Siegrist for *worniu-GAL4* and *UAS-dp60*).

4) Is there any long-term consequence of reduced neuroblast proliferation? Or is the process just delayed by sleep deprivation? Can this be addressed or discussed?

We thank the reviewers for this question, as it is one of our major long-term interests. The relatively brief sleep loss protocol described in this manuscript is unlikely to have long-lasting permanent consequences. We are actively pursuing approaches to achieve prolonged sleep deprivation in second instar larvae, with the goal of understanding how a more severe reduction in NB proliferation during this time window affects structural development of the brain, as well as later behaviors. We now address these issues in the Discussion section.

5)The study established sleep using many established criteria: distinct sleep posture, arousal threshold, and sleep rebound after deprivation. However, it does not mention sleep pressure. A revised manuscript should provide an explanation as to why.

We thank the reviewers for this comment, which motivated experiments we now show in Figure 3—figure supplement 2. Following sleep deprivation, larvae exhibit increased sleep depth (more difficult to arouse) in addition to increased sleep quantity.

6) Subsection “Octopamine controls sleep-wake in larvae”: It seems too strong to state that octopamine "controls sleep-wake in larvae." Modulates? Influences?

We have changed the text accordingly.

7) Subsection “Octopamine controls sleep-wake in larvae”: It is suggested, based on a reference to the Aston-Jones and Bloom paper, that norepinephrine plays "a conserved role in developmental sleep across species." Leaving aside the awkward sentence, the reference is to an adult rat sleep paper. I don't know of any studies on norepinephrine and sleep in early development in rats or any other species, but if they exist they should be cited here. Regardless, suggestions of conserved functions at this early juncture seem premature.

We thank the reviewers for this point and have removed this sentence.

8) The reviewers agree that the current analyses are perfectly adequate for the current manuscript. However, here are two possibly helpful suggestions for future manuscripts.a) The choice of 6 seconds between video frames is longer than ideal and the authors are encouraged to reconsider this choice in the future. The average sleep bout lengths of 15-30 seconds represent only 2.5-5 frames, so 6 seconds provides a course measure of bout length; preferably, sampling intervals should be smaller in relation to bout lengths. Also, with more precise measurement of bout lengths, it will be possible to analyze the statistical distributions of the sleep and wake bout lengths, as has been done profitably in infant rats and the adults of several mammalian species. For example, do the sleep bouts distribute exponentially, as has been found in mammalian sleep? More precise measurement would also allow the authors to more sensitively assess bout consolidation, developmental changes, etc. Such analyses can also provide clues to mechanism.b) A related issue concerns the definition of larval sleep as a minimum of 6 seconds of quiescence, which is defined as no evidence of activity across two frames. This definition, though clear, appears to be an arbitrary one. It also makes the comparison to the 6-second definition of sleep in zebrafish (Discussion section) less interesting. Why not measure sleep postures more accurately and analyze the frequency distributions of sleep times with greater precision (e.g., 1-second bins)?

We thank the reviewers for these points and agree that more frequent sampling (as in Figure 2) will be informative going forward.